chemical biology

micro-arc oxidation, titanium, MG63, osteogenic differentiation, dental implants

**Authors for correspondence:**
Min Qi
e-mail: minqi@dlut.edu.cn
Yi Liu
e-mail: liuyi@cmu.edu.cn

This article has been edited by the Royal Society of Chemistry, including the commissioning, peer review process and editorial aspects up to the point of acceptance.

# Micro/nano-hierarchical structured TiO₂ coating on titanium by micro-arc oxidation enhances osteoblast adhesion and differentiation

Xumeng Pan[1], Yada Li[2], Adil O. Abdullah[1], Weiqiang Wang[2], Min Qi[2] and Yi Liu[1]

[1]School of Stomatology, China Medical University, Shenyang 110013, People's Republic of China
[2]School of Materials Science and Engineering, Dalian University of Technology, Dalian 116024, People's Republic of China

XP, 0000-0001-8181-8936; MQ, 0000-0001-6073-3765; YL, 0000-0002-4640-3396

Nano-structured and micro/nano-hierarchical structured TiO₂ coatings were produced on polished titanium by the micro-arc oxidation (MAO) technique. This study was conducted to screen a suitable structured TiO₂ coating for osteoblast adhesion and differentiation in dental implants. The formulation was characterized by scanning electron microscopy (SEM), X-ray diffraction (XRD) and wettability testing. Adhesion, proliferation and osteogenic differentiation of MG63 cells were analysed by SEM, Cell Counting Kit-8 (CCK-8) and quantitative real-time PCR. The micro/nano-hierarchical structured TiO₂ coating with both slots and pores showed the best morphology and wettability. XRD analysis revealed that rutile predominated along with a minor amount of anatase in both TiO₂ coatings. Adhesion and extension of MG63 cells on the micro/nano-hierarchical structured TiO₂ coating were the most favourable. MG63 cells showed higher growth rates on the micro/nano-hierarchical structured TiO₂ coating at 1 and 3 days. Osteogenic-related gene expression was markedly increased in the micro/nano-hierarchical structured TiO₂ coating group compared with the polished titanium group at 7, 14 and 21 days. These results revealed the micro/nano-hierarchical structured TiO₂ coating as a promising surface modification and suitable biomaterial for use with dental implants.

# 1. Introduction

Osseointegration is defined as the direct bonding of living bone tissue with surgical implants that can replace bone and perform load-bearing functions [1]. Osteoblast adhesion and differentiation on implant surfaces are two important indices affecting osseointegration [2]. Bone formation mainly depends on the surface characteristics of surgical implants [3]. A rough surface is beneficial for osteoblast differentiation, as a rough and porous surface structure can enlarge the contact area between the material and osteoblast, promoting osteoblast stretching and growth into the porous surface [4]. Moreover, the surface chemical properties of implants are critical for early bone formation [5]. Enhanced surface energy and wettability can stimulate the interaction between the implant surface and its surrounding biological environment. A hydroxylated or hydrated surface, which exhibits immediate wettability, contributes to the production of a more differentiated osteoblast phenotype [6].

Titanium (Ti) and its alloys have been widely applied in the field of dental implants for several years. During exposure to the air, a complete and dense $TiO_2$ layer forms spontaneously on the surface of Ti [7]. This oxide layer leads to remarkable corrosion resistance; on the other hand, this inherent oxide layer results in the growth of fibrous connective tissue, preventing new bone formation on the implant surface, which retards osseointegration between the implant and bone. Thus, surface modifications that avoid these side effects are necessary for applying Ti in dental implants [1,8].

Micro-arc oxidation (MAO), derived from anodization, transforms amorphous oxide into a crystalline phase by flash sintering from high-temperature and high-voltage area of micro-plasma [9]. Firmly adherent porous and rough coatings can be formed on the surface of Ti by MAO [10], and these surfaces have been shown to be suitable for osseointegration [11,12].

In this study, MAO treatments were performed as follows: Ti substrates in one group were prepared under conditions of 465 V, 600 Hz and 9% duty circle in electrolyte solution of $0.1 \, \text{mol} \, l^{-1}$ $Li_2B_4O_7$ for 2 min to form a nano-structured $TiO_2$ coating on the surface; substrates in the other group were prepared under conditions of 465 V, 600 Hz and 11% duty circle in electrolyte solution of $0.1 \, \text{mol} \, l^{-1}$ $Li_2B_4O_7$ for 13 min to prepare a micro/nano-structured $TiO_2$ coating on the surface. The surface characteristics of nano-structured and micro/nano-structured $TiO_2$ coatings applied by MAO on the Ti surface were compared. MG63 osteoblast-like cells were used to evaluate cell adhesion, proliferation and osteogenic differentiation. This study was conducted to screen for a suitable structured $TiO_2$ coating on Ti as a surface modification that can be used in the field of dental implants.

# 2. Material and methods

## 2.1. Specimen preparation

Commercial pure titanium (cp-Ti, TA-2) was used as the substrate material. Ti plates were cut to a size of $10 \times 10 \times 1$ mm. All substrates were mechanically polished sequentially with SiC papers of grit 180, 400, 800 and 1000, and then ultrasonically cleaned sequentially with acetone, absolute ethyl alcohol and deionized water, followed by air-drying. MAO coatings were prepared by using a DC pulse power supply (Harbin Institute of Technology, China). The Ti substrate was used as the anode and the stainless steel electrolyser was used as the cathode. MAO treatments were performed as follows: Ti substrates in one group were prepared under conditions of 465 V, 600 Hz and 9% duty circle in electrolyte solution of $0.1 \, \text{mol} \, l^{-1}$ $Li_2B_4O_7$ for 2 min to form a nano-structured coating on the surface; this group was named 9%-2MAO. Substrates in the other group were prepared under conditions of 465 V, 600 Hz and 11% duty circle in electrolyte solution of $0.1 \, \text{mol} \, l^{-1}$ $Li_2B_4O_7$ for 13 min to prepare a micro/nano-structured coating on the surface; this group was named 11%-13MAO. Mechanically polished Ti substrates were used as the control group. All samples were cleaned with deionized water, air-dried and then autoclaved for 30 min.

## 2.2. Surface characterization

### 2.2.1. Surface morphology and phase composition

The surface morphology and phase composition of Ti, 9%-2MAO and 11%-13MAO were assessed by field emission scanning electron microscopy (FE-SEM, Zeiss, Germany) and analysed by X-ray diffraction (XRD, Empyrean, The Netherlands), respectively.

### 2.2.2. Wettability

Contact angles were measured to assess the hydrophobicity or hydrophilicity of the sample surfaces. Two microlitres of deionized water were added on to the sample surface, and the contact angle was measured with a DSA100 optical contact angle system (Krüss Scientific, Germany). Images were captured and transferred to the computer, and contact angles were measured and determined from the images.

## 2.3. Biological analysis

### 2.3.1. Cell lines and cell culture

All experiments were performed by using the MG63 osteoblast-like cell line (kindly provided by College of Stomatology, Dalian Medical University). The cells were maintained at 37°C in a fully humidified incubator with 5% $CO_2$ in Dulbecco's modified Eagle's medium (DMEM, Invitrogen, USA) containing 10% fetal bovine serum (FBS, Gibco BRL, USA). All media containing serum were changed every other day.

### 2.3.2. Cell adhesion and morphology

Ti, 9%-2MAO and 11%-13MAO were placed in 24-well plates prior to cell seeding, and 1 ml of cell suspension containing $2 \times 10^4$ cells was seeded onto the surface of each sample and incubated for 2, 6 and 24 h. At each time point, the samples were transferred to a new 24-well plate and washed with phosphate-buffered saline (PBS) and fixed with 2.5% glutaraldehyde in PBS for 4 h. The samples were then dehydrated with a graded series of ethanol (30, 50, 75, 95 and 100%), dried, and sputter-coated with gold. Cell morphologies on the samples were examined by SEM.

### 2.3.3. Cell proliferation assay

Cell proliferation was measured using Cell Counting Kit-8 (CCK-8, Takara, Japan) according to the manufacturer's instructions. Samples were added to 24-well plates and cells were seeded onto each sample at a density of $2 \times 10^4$ cells, followed by incubation for 1, 3, 5 and 7 days. At predetermined time points, samples were transferred to a new 24-well plate and washed with PBS. Cells on the samples were incubated with 300 µl of DMEM and 30 µl of CCK-8 solution for 1 h in the incubator, and then optical density (OD) was measured using a microplate reader at a wavelength of 450 nm.

### 2.3.4. RNA isolation and quantitative real-time PCR

Samples were added to a 24-well plate; $3.5 \times 10^4$ cells were dispensed onto each sample and cultured for 7, 14 and 21 days. Cells on each disc were lysed with Trizol reagent (Invitrogen, USA) and the lysates were collected by pipetting and centrifugation. Total RNAs were extracted using RNAiso Plus reagent (TaKaRa, Japan) according to the manufacturer's instructions. First-strand complementary DNA (cDNA) was generated from mRNA by using PrimerScript$^{TM}$ RT reagent (TaKaRa, Japan). Quantitative real-time PCR was performed using SYBR® Premix ExTaq$^{TM}$ II (TaKaRa, Japan) on a Bio-Rad iQTM5 system (Hercules, USA). Individual gene expression levels were normalized to GAPDH expression. The oligonucleotide primers used in the amplification reaction were 5′-GCTTGGTCCACTTGCTTGAAGA-3′ and 5′-GAGCATTGCCTTTGATTGCTG-3′ for collagen type I-α1 (COLI-α1); 5′-GGAACGGACATTCGGTCCT-3′ and 5′-GGAAGCAGCAACGCTAGAAG-3′ for bone morphogenetic protein 2 (BMP2); 5′-GACGAGTTGGCTGACCACA-3′ and 5′-CAAGGGGAAGAG-GAAAGAAGG-3′ for osteocalcin (OCN); and 5′-GCACCGTCAAGGCTGAGAAC-3′ and 5′-TGGTGAAGACGCCAGTGGA-3′ for GAPDH.

## 2.4. Statistical analysis

The data were presented as the mean ± s.d. and analysed with SPSS 17.0 software (SPSS, Inc., USA) by one-way analysis of variance (ANOVA). The level of statistical significance was defined at $p \leq 0.05$. All experiments were performed in triplicate.

# 3. Results

## 3.1. Material characterization

The microstructure was characterized by SEM in the Ti, 9%-2MAO and 11%-13MAO groups, as shown in figure 1. The Ti surface was relatively smooth with parallel scratches aligned along the grinding direction (figure 1A). The 9%-2MAO surface showed a typical porous structure at the nano-scale, and nanopores 10–300 nm in diameter were formed on the coating surface (figure 1B). Micro-sized slots (3–7 μm) and nano-sized pores (diameter: 10–300 nm) were equally distributed and linked with each other on the surface of 11%-13MAO (figure 1C). The surface phase structure was analysed by XRD (figure 2). Stable rutile and metastable anatase were clearly observed in the oxide coatings of 9%-2MAO and 11%-13MAO, and the rutile peaks increased gradually with an increased MAO treatment time. Wettability was tested in a contact angle assay (figure 3). The static water contact angles on the surfaces of 9%-2MAO ($22.97 \pm 2.358°$) (figure 3b) and 11%-13MAO ($5.44 \pm 0.813°$) (figure 3c) were significantly lower than that of Ti ($82.30 \pm 3.301°$) (figure 3a) ($p < 0.05$). Moreover, 11%-13MAO showed the lowest contact angle.

## 3.2. Micro/nano-hierarchical structured TiO$_2$ coating promoted cell adhesion and proliferation

The adhesion of MG63 cells on the Ti, 9%-2MAO and 11%-13MAO surfaces was examined by SEM after incubation for 2, 6 and 24 h (figure 4). In the 11%-13MAO group, the cell shape was polygonal and more extended than in the other two groups at the beginning of 2 h culture, whereas MG63 cells on the Ti and 9%-2MAO surfaces remained spherical.

The proliferation of MG63 cells on the Ti, 9%-2MAO and 11%-13MAO surfaces was investigated at 1, 3, 5 and 7 days (figure 5). On days 1 and 3, cell proliferation was significantly increased in the 11%-13MAO group compared with the Ti group ($p < 0.05$). Increased cell proliferation was observed in the 9%-2MAO group on day 5 compared with the Ti group ($p < 0.05$).

## 3.3. Micro/nano-hierarchical structured TiO$_2$ coating increased cellular osteogenic differentiation

The expression levels of the cellular osteogenic-related markers COLI-α1, BMP2 and OCN were determined in each group after cell culture for 7, 14 and 21 days to assess MG63 osteogenic differentiation (figure 6). On day 7, the expression levels of all osteogenic-related markers in the 9%-2MAO and 11%-13MAO groups were significantly higher than those in the polished Ti group. Among the three groups, the highest COLI-α1, BMP2 and OCN expression was observed in the 11%-13MAO group ($p < 0.05$), indicating the 11%-13MAO promoted MG63 osteogenic differentiation.

# 4. Discussion

Ti is used for dental implants, but its further applications are limited because of its ineffectiveness for osseointegration. MAO, a synergic process in thermochemistry, plasma chemistry and electrochemistry, has been proposed for modifying biological materials, such as dental implants [11]. Liu *et al.* [13] prepared a unique 'cortex-like' micro/nano-structured TiO$_2$ coating with micrometre-scale slots and nano-scale pores by MAO technology. Based on *in vitro* and *in vivo* experiments, Li *et al.* [14] suggested that the 'cortex-like' dual-scale structure could improve implant success. This micro/nano-structure is significantly superior to micro-structures in terms of cytocompatibility and osseointegration and significantly increased osteoblast adhesion and differentiation. Nanoporous TiO$_2$ promoted osteoblast attachment and proliferation [15]. In the current study, we prepared nano-structured and micro/nano-hierarchical structured TiO$_2$ coatings by MAO, using cp-Ti as the control group, to screen for promising TiO$_2$ structures useful in dental implant therapy.

The surface characteristics of our prepared TiO$_2$ coatings were evaluated by SEM and XRD. SEM analysis revealed that 9%-2MAO had a nanoporous structure, while 11%-13MAO consisted of equally distributed micro-sized slots (3–7 μm wide) and nano-sized pores (diameter: 10–300 nm) and showed a typical micro/nano dual structure that mimicked the hierarchical characteristics of bone. The surface phase structure was analysed by XRD. Nano-structured and micro/nano-hierarchical structured TiO$_2$ coatings on the Ti surface were mainly rutile and contained a small amount of anatase TiO$_2$. This was because the voltage and reaction temperature of MAO treatment were both high. The 9%-2MAO and

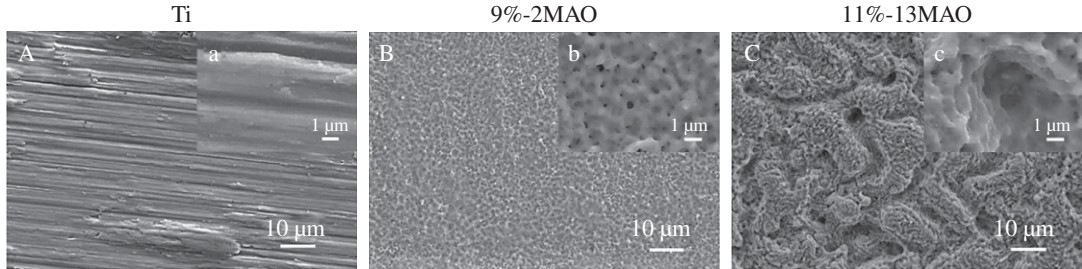

**Figure 1.** SEM images of Ti (A, a), 9%-2MAO (B, b) and 11%-13MAO (C, c).

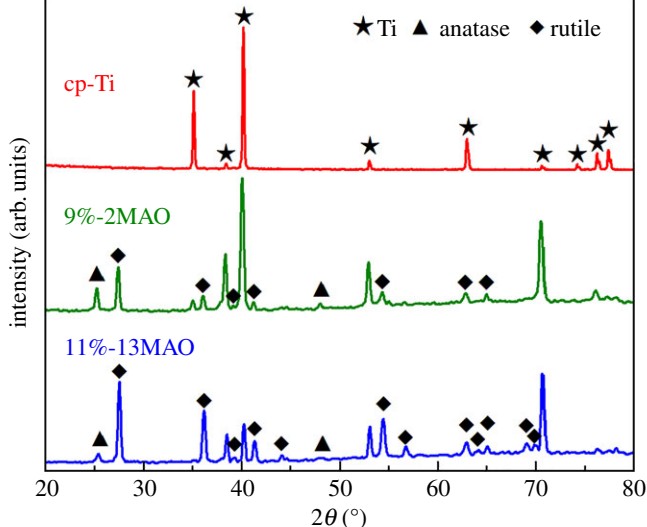

**Figure 2.** XRD patterns of cp-Ti, 9%-2MAO and 11%-13MAO.

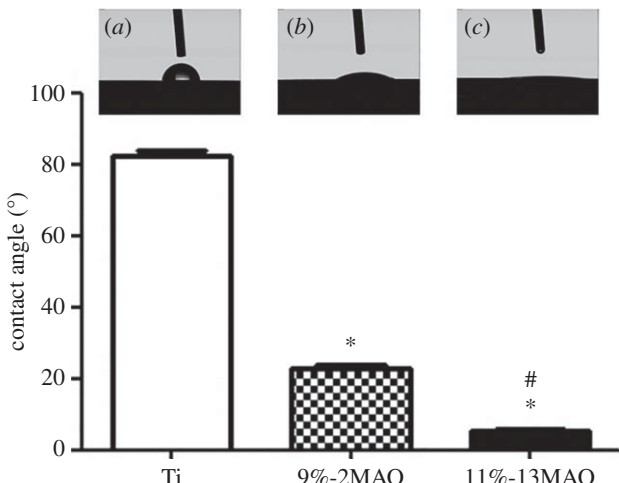

**Figure 3.** Contact angles of Ti (a), 9%-2MAO (b) and 11%-13MAO (c) with respect to 2 µl of deionized water. Asterisk (*) means $p < 0.05$ compared with Ti group; # means $p < 0.05$ compared with 9%-2MAO group.

11%-13MAO coatings dominated at high-temperature stable phase rutile. Anatase $TiO_2$ shows better biocompatibility and bioactivity, which may induce the formation of bone-like apatite [10]. The duration of MAO was very short in the 9%-2MAO group, and the thinner oxide coating led to a clear Ti peak for the substrate. The static water contact angle was measured to quantify wettability. The boundary angle of 65° is used to distinguish a hydrophilic from a hydrophobic surface [16].

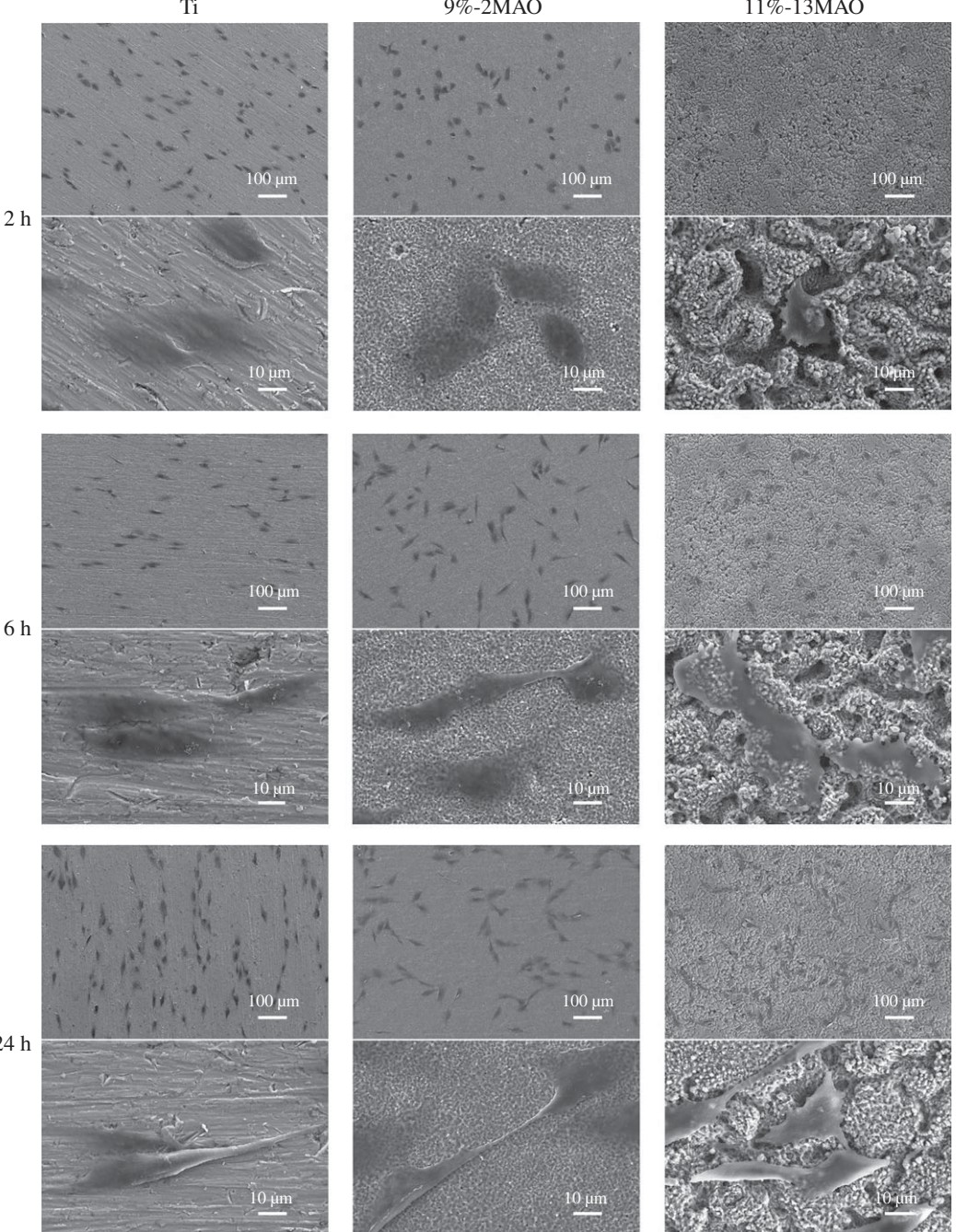

**Figure 4.** SEM images showing surface morphologies of MG63 cells cultured on Ti, 9%-2MAO and 11%-13MAO at 2, 6 and 24 h.

Our study and previous studies showed that the Ti surface was hydrophobic. However, after MAO treatment, the oxide surface was hydrophilic. The 11%-13MAO surface was super-hydrophilic. Numerous studies have suggested that a hydrophilic surface tends to increase biological activity [17], cell adhesion, differentiation [18] and bone mineralization [16]. The surface morphology of implants is another crucial point affecting the adhesion and differentiation of osteoblasts during the initial phase of osseointegration and long-term bone remodelling [19,20]. The micro-nano dual structure exhibited rough surface characteristic and favourable wettability, which enhanced hydrophilicity and cell adhesion [21,22]. Therefore, the biological properties of the MAO coatings were further analysed.

MG63 osteosarcoma cell lines are commonly used in osteogenic differentiation analysis [23–25]. Our results showed that the micro/nano-hierarchical structured TiO$_2$ coating applied by MAO contributed to osteoblast adhesion. This is consistent with the results of Zhang *et al.* [26], who plated SaOS-2 osteosarcoma cells on macro/mesoporous-structured coatings to evaluate the initial adhesion,

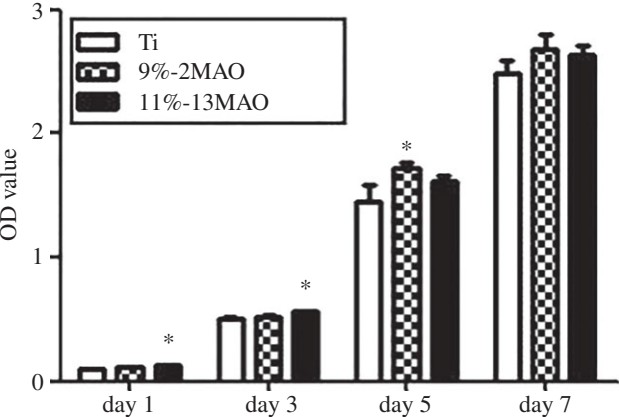

**Figure 5.** Proliferation of MG63 cells grown on Ti, 9%-2MAO and 11%-13MAO surfaces at 1, 3, 5 and 7 days. Asterisk (*) means $p < 0.05$ compared with Ti group.

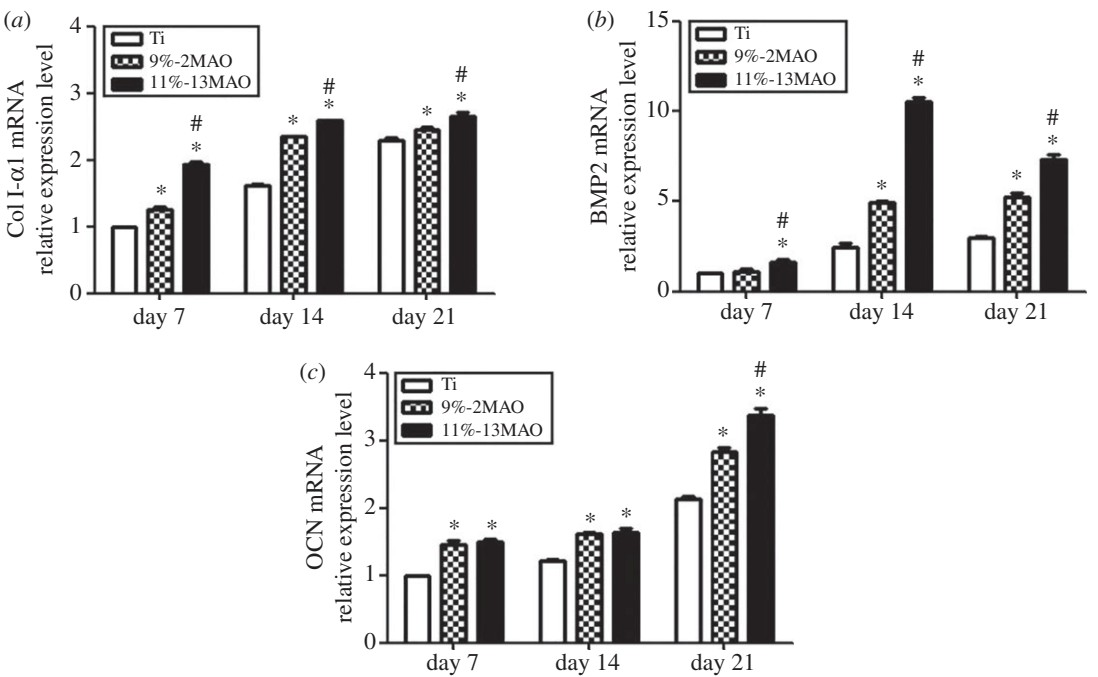

**Figure 6.** Gene expression of COLI-α1 (*a*), BMP2 (*b*) and OCN (*c*) in MG63 cells on Ti, 9%-2MAO and 11%-13MAO surfaces at 7, 14 and 21 days. Asterisk (*) means $p < 0.05$ compared with Ti group; # means $p < 0.05$ compared with 9%-2MAO group.

proliferation and differentiation of cells. They showed that the hierarchical structure is a promising surface morphology for implants.

Osteoblast proliferation and differentiation on $TiO_2$ coatings were further evaluated. As described above, the osteogenic differentiation of cells surrounding the implant is essential for achieving better osseointegration [27]. Extracellular matrix (ECM) proteins such as COLI-α1, BMP2 and OCN were measured to evaluate the level of osteogenic differentiation. COLI-α1 and BMP2 are early markers of osteogenic differentiation [28–30], whereas OCN is a marker of the late stage [31]. At days 1 and 3, compared with Ti, the micro/nano-hierarchical structured $TiO_2$ coating significantly promoted MG63 cell proliferation. Starting on day 5, the difference was no longer significant. Subsequently, cells on the 11%-13MAO surface participated in osteogenic differentiation, and the relative mRNA expression levels of COLI-α1, BMP2 and OCN were all clearly elevated. Previous studies [19,32–35] showed that the surface morphology of materials influences the adhesion and differentiation of osteoblasts throughout the process from the initial phase of osseointegration to subsequent bone remodelling. Thus, our prepared micro/nano-hierarchical structured $TiO_2$ coating showed positive effects on both early proliferation and differentiation of osteoblasts. Moreover, BMP2 is not only an osteogenic

marker, but also a critical regulator of cell adhesion [36]. BMP2 expression was highest in the 11%-13MAO group. Tight cell adhesion and pile-up are two key factors in promoting osteogenesis [37]. This explains why our prepared micro/nano-hierarchical structured $TiO_2$ coating promoted both osteoblast proliferation and differentiation. Further studies are needed to determine the effect of MAO times or duty circles on the structure of the $TiO_2$ coating. The rate and quality of osseointegration *in vivo* requires further exploration.

# 5. Conclusion

In summary, our prepared nano-structured and micro/nano-hierarchical structured $TiO_2$ coatings on the Ti surface by MAO significantly increased hydrophilicity as well as promoted osteoblast adhesion and differentiation. The micro/nano-hierarchical structured $TiO_2$ coating showed the most suitable properties and may be useful for application to Ti for dental implants.

Ethics. The research project was reviewed by the Medical Ethics Committee of our hospital and complied with the requirements of the Helsinki Declaration and related medical ethics.

Data accessibility. Data available from the Dryad Digital Repository at: https://doi.org/10.5061/dryad.bj3570d [38].

Authors' contributions. X.P. performed the experiments, analysed data and wrote the manuscript. Y.L. constructed samples and carried out part of the experiments. A.O.A. revised the manuscript. W.W. critically reviewed the manuscript. Y.L. and M.Q. conceived, designed and supervised the whole experiments.

Competing interests. The authors declare no competing financial interest.

Funding. This research was supported by the National Natural Science Foundation of China (grant no. 51371042), Liaoning Province Natural Science Fund Project (grant no. 20180550420) and Co-professional Graduates Program of the Liaoning Provincial Universities (grant no. 115-3110617005).

Acknowledgements. The authors also appreciated the kind help from Prof. Jing Xiao and Prof. Weidong Niu in College of Stomatology, Dalian Medical University during cell culture.

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
