## [Reviewer comments · Royal Society Open Science]

Review History

RSOS-182031.R0 (Original submission)

Review form: Reviewer 1

Is the manuscript scientifically sound in its present form?

Yes

Are the interpretations and conclusions justified by the results?

Yes

Is the language acceptable?

Yes

Is it clear how to access all supporting data?

Yes

Do you have any ethical concerns with this paper?

No

Have you any concerns about statistical analyses in this paper?

Yes

Recommendation?

Accept with minor revision (please list in comments)

Comments to the Author(s)

The paper "Optimized micro/nano-hierarchical structured TiO₂ coating on titanium by micro-arc oxidation enhances osteoblast adhesion and differentiation" describes the preparation and physico-chemical and biological characterization of micro/nano structured TiO₂ coatings by micro-arc oxidation, making this paper of interest for broader public.

This study is well conceived and results clearly presented, however there are several points that should be addressed before accepting the paper:

a) Introduction

- More details about TiO₂ coatings obtained by micro-arc oxidation should be given.

b) Statistical analysis

- The number of repetitions is not given, although it can be deduced from SI data.

- How were proliferation data Fig. 5 treated considering that they consist of both repeat and replicate measurements?

c) Material characterization

- In Fig 1. higher magnification micrographs for Ti and 11%-13 MAO should be provided.

- In Fig. 2. XRD of Ti should be provided.

Review form: Reviewer 2

Is the manuscript scientifically sound in its present form?

No

Are the interpretations and conclusions justified by the results?

No

Is the language acceptable?

No

Is it clear how to access all supporting data?

Yes

Do you have any ethical concerns with this paper?

No

Have you any concerns about statistical analyses in this paper?

No

Recommendation?

Major revision is needed (please make suggestions in comments)

Comments to the Author(s)

This manuscript, entitled, 'Optimized micro/nano-hierarchical structured TiO₂ coating on titanium by micro-arc oxidation enhances osteoblast adhesion and differentiation,' is considered to be interesting and to be within the scope of this journal. However, there are many concerns to be addressed. And English editing is necessary.

Introduction

Line 51, page 2: Moreover, the surface...bone formation.

References are needed.

Line 4, page 3: Hydroxylated or hydrated...differentiated osteoblast phenotype.

References are needed.

Line 18, page 3: The presence of TiO₂ layer...Ti in dental implants.

The reviewer disagrees with the authors. If the authors really believe that this description is right, please, explain it in detail, including proper references.

Line 35, page 3: Micro-arc oxidation (MAO)...of micro-plasma.

References are needed.

Line 42, page 3: Firmly adherent porous...surfaces for osseointegration [7, 8].

What is the coating material formed on the surface of Ti by MAO? The reviewer thinks that MAO produces a TiO₂ layer. Why is this layer different from that preventing osseointegration? Just the change of crystallinity is not considered to be the transformation from being biotoxic to being biocompatible.

Materials and methods

2.1. Specimen preparation

Which grade of cp-Ti was used in this study?

Results

3.1. Material characterization

Line 46, page 8: The surface of Ti was primarily smooth.

The authors are asked for more detailed description of the cp-Ti surface. Why did the authors think that the Fig. 1A image showed a smooth surface? What does the word, 'smooth,' mean exactly?

Line 4, page 9: The surface chemistry was analyzed by XRD (Fig. 2)

Although XRD has some information in surface chemistry, XRD is usually known to analyze a surface in crystallinity. The authors analyzed the surfaces in phase only, not in chemical elements or in the percentages (atomic or weight) of the elements in this study. Maybe, oxygen is also detected on the titanium surfaces if the authors analyze the surfaces by X-ray photospectroscopy or energy dispersive spectroscopy.

Also, the significant figures in the results are recommended to be consistent.

Discussion

Line 21, page 14: Here, we aimed...dental implants therapy.

The micro/nano-hierarchical structure of the authors is not considered to be an optimized biomaterial, based on the results of this study. The authors did not perform the experiments depending on the MAO times or the duty circles. The results were not statistically analyzed when these factors were independent variables. The word, 'optimized,' is inadequate in this study. Further studies are definitely needed.

Please, describe the limitations of this study. The authors did not perform in vivo experiments. What is the difference between the authors' study and the previous study of Li YD, et al (Reference No. 10), which includes the in vivo results? The reviewer recommends that the authors should describe improvement of this study, compared with the previous one.

Decision letter (RSOS-182031.R0)

06-Feb-2019

Dear Dr Liu:

Title: Optimized micro/nano-hierarchical structured TiO₂ coating on titanium by micro-arc oxidation enhances osteoblast adhesion and differentiation
Manuscript ID: RSOS-182031

The editor assigned to your manuscript has now received comments from reviewers. We would like you to revise your paper in accordance with the referee and Subject Editor suggestions which can be found below (not including confidential reports to the Editor). Please note this decision does not guarantee eventual acceptance.

Please submit your revised paper before 01-Mar-2019. Please note that the revision deadline will expire at 00.00am on this date. If we do not hear from you within this time then it will be assumed that the paper has been withdrawn. In exceptional circumstances, extensions may be possible if agreed with the Editorial Office in advance. We do not allow multiple rounds of revision so we urge you to make every effort to fully address all of the comments at this stage. If deemed necessary by the Editors, your manuscript will be sent back to one or more of the original reviewers for assessment. If the original reviewers are not available we may invite new reviewers.

Please also include the following statements alongside the other end statements. As we cannot publish your manuscript without these end statements included, if you feel that a given heading is not relevant to your paper, please nevertheless include the heading and explicitly state that it is not relevant to your work.

- Ethics statement

Please clarify whether you received ethical approval from a local ethics committee to carry out your study. If so please include details of this, including the name of the committee that gave consent in a Research Ethics section after your main text. Please also clarify whether you received informed consent for the participants to participate in the study and state this in your Research Ethics section.

OR

Please clarify whether you obtained the necessary licences and approvals from your institutional animal ethics committee before conducting your research. Please provide details of these licences and approvals in an Animal Ethics section after your main text.

OR

Please clarify whether you obtained the appropriate permissions and licences to conduct the fieldwork detailed in your study. Please provide details of these in your methods section.

RSC Associate Editor:
Comments to the Author:
(There are no comments.)

RSC Subject Editor:
Comments to the Author:
(There are no comments.)

Reviewers' Comments to Author:
Reviewer: 1

Comments to the Author(s)

The paper "Optimized micro/nano-hierarchical structured TiO₂ coating on titanium by micro-arc oxidation enhances osteoblast adhesion and differentiation" describes the preparation and physico-chemical and biological characterization of micro/nano structured TiO₂ coatings by micro-arc oxidation, making this paper of interest for broader public.

This study is well conceived and results clearly presented, however there are several points that should be addressed before accepting the paper:

a) Introduction

- More details about TiO₂ coatings obtained by micro-arc oxidation should be given.

b) Statistical analysis

- The number of repetitions is not given, although it can be deduced from SI data.

- How were proliferation data Fig. 5 treated considering that they consist of both repeat and replicate measurements?

c) Material characterization

- In Fig 1. higher magnification micrographs for Ti and 11%-13 MAO should be provided.
- In Fig. 2. XRD of Ti should be provided.

Reviewer: 2

Comments to the Author(s)

This manuscript, entitled, 'Optimized micro/nano-hierarchical structured TiO₂ coating on titanium by micro-arc oxidation enhances osteoblast adhesion and differentiation,' is considered to be interesting and to be within the scope of this journal. However, there are many concerns to be addressed. And English editing is necessary.

Introduction

Line 51, page 2: Moreover, the surface...bone formation.

References are needed.

Line 4, page 3: Hydroxylated or hydrated...differentiated osteoblast phenotype.

References are needed.

Line 18, page 3: The presence of TiO₂ layer...Ti in dental implants.

The reviewer disagrees with the authors. If the authors really believe that this description is right, please, explain it in detail, including proper references.

Line 35, page 3: Micro-arc oxidation (MAO)...of micro-plasma.

References are needed.

Line 42, page 3: Firmly adherent porous...surfaces for osseointegration [7, 8].

What is the coating material formed on the surface of Ti by MAO? The reviewer thinks that MAO produces a TiO₂ layer. Why is this layer different from that preventing osseointegration? Just the change of crystallinity is not considered to be the transformation from being biotoxic to being biocompatible.

Materials and methods

2.1. Specimen preparation

Which grade of cp-Ti was used in this study?

Results

3.1. Material characterization

Line 46, page 8: The surface of Ti was primarily smooth.

The authors are asked for more detailed description of the cp-Ti surface. Why did the authors think that the Fig. 1A image showed a smooth surface? What does the word, 'smooth,' mean exactly?

Line 4, page 9: The surface chemistry was analyzed by XRD (Fig. 2)

Although XRD has some information in surface chemistry, XRD is usually known to analyze a surface in crystallinity. The authors analyzed the surfaces in phase only, not in chemical elements or in the percentages (atomic or weight) of the elements in this study. Maybe, oxygen is also detected on the titanium surfaces if the authors analyze the surfaces by X-ray photospectroscopy or energy dispersive spectroscopy.

Also, the significant figures in the results are recommended to be consistent.

Discussion

Line 21, page 14: Here, we aimed...dental implants therapy.

The micro/nano-hierarchical structure of the authors is not considered to be an optimized biomaterial, based on the results of this study. The authors did not perform the experiments depending on the MAO times or the duty circles. The results were not statistically analyzed when these factors were independent variables. The word, 'optimized,' is inadequate in this study. Further studies are definitely needed.

Please, describe the limitations of this study. The authors did not perform in vivo experiments. What is the difference between the authors' study and the previous study of Li YD, et al (Reference No. 10), which includes the in vivo results? The reviewer recommends that the authors should describe improvement of this study, compared with the previous one.

Author's Response to Decision Letter for (RSOS-182031.R0)

See Appendix A.

RSOS-182031.R1 (Revision)

Review form: Reviewer 1

Is the manuscript scientifically sound in its present form?

Yes

Are the interpretations and conclusions justified by the results?

Yes

Is the language acceptable?

Yes

Is it clear how to access all supporting data?

Yes

Do you have any ethical concerns with this paper?

No

Have you any concerns about statistical analyses in this paper?

No

Recommendation?

Accept as is

Comments to the Author(s)

The authors have addressed all remarks satisfactorily.

Review form: Reviewer 2

Is the manuscript scientifically sound in its present form?

Yes

Are the interpretations and conclusions justified by the results?

Yes

Is the language acceptable?

Yes

Is it clear how to access all supporting data?

Yes

Do you have any ethical concerns with this paper?

No

Have you any concerns about statistical analyses in this paper?

No

Recommendation?

Accept as is

Comments to the Author(s)

This revised manuscript is considered to be well-edited according to the reviewer's comments.

Decision letter (RSOS-182031.R1)

12-Mar-2019

Dear Dr Liu:

Title: Micro/nano-hierarchical structured TiO₂ coating on titanium by micro-arc oxidation enhances osteoblast adhesion and differentiation

Manuscript ID: RSOS-182031.R1

It is a pleasure to accept your manuscript in its current form for publication in Royal Society Open Science. The chemistry content of Royal Society Open Science is published in collaboration with the Royal Society of Chemistry.

Yours sincerely,

Dr Laura Smith

Publishing Editor, Journals

RSC Associate Editor:
Comments to the Author:
(There are no comments.)

RSC Subject Editor:
Comments to the Author:
(There are no comments.)

Reviewer(s)' Comments to Author:
Reviewer: 2

Comments to the Author(s)
This revised manuscript is considered to be well-edited according to the reviewer's comments.

Reviewer: 1

Comments to the Author(s)
The authors have addressed all remarks satisfactorily.

Appendix A

Response to Referees

Reviewers' Comments to Author:

Reviewer: 1

a) Introduction

- More details about TiO₂ coatings obtained by micro-arc oxidation should be given.

Response : I have given more details about TiO₂ coatings obtained by micro-arc oxidation in the last paragraph of the Introduction, marked in red.

b) Statistical analysis

- The number of repetitions is not given, although it can be deduced from SI data.

Response : All experiments were performed in triplicate. And I have added it in the part of "2.4. Statistical analysis", marked in red.

- How were proliferation data Fig. 5 treated considering that they consist of both repeat and replicate measurements?

Response : The experiments of cell proliferation were performed in triplicate. Three replicate wells were set for each group in each experiment, and the mean value of three replicate wells was taken, so three averages per group can be given. One-way analysis of variance (ANOVA) was performed using three averages.

c) Material characterization

- In Fig 1. higher magnification micrographs for Ti and 11%-13 MAO should be provided.

Response : I have provided higher magnification micrographs for Ti and 11%-13 MAO in Fig 1.

- In Fig. 2. XRD of Ti should be provided.

Response : I have provided XRD of Ti in Fig 2.

Reviewer: 2

Response : This is the certificate of English editing.

CERTIFICATE OF ENGLISH EDITING

This document certifies that the paper listed below has been edited to ensure that the language is clear and free of errors. The logical presentation of ideas and the structure of the paper were also checked during the editing process. The edit was performed by professional editors at Editage, a division of Cactus Communications. The intent of the author's message was not altered in any way during the editing process. The quality of the edit has been guaranteed, with the assumption that our suggested changes have been accepted and have not been further altered without the knowledge of our editors.

TITLE OF THE PAPER

Micro/nano-hierarchical structured TiO₂ coating on titanium by micro-arc oxidation enhances osteoblast adhesion and differentiation

AUTHORS

Xumeng Pan, Yada Li, Adil O. Abdullah, Weiqiang Wang, Min Qi,* Yi Liu,*

JOB CODE

PXPXM_1

Signature

Vikas Narang

Vikas Narang,
Senior Vice President,
Operations-Author Services, Editage

Date of Issue
February 25, 2019

Editage, a brand of Cactus Communications, offers professional English language editing and publication support services to authors engaged in over 500 areas of research. Through its community of experienced editors, which includes doctors, engineers, published scientists, and researchers with peer review experience, Editage has successfully helped authors get published in internationally reputed journals. Authors who work with Editage are guaranteed excellent language quality and timely delivery.

CACTUS

Contact Editage

Worldwide request@editage.com +1 877-334-8243 www.editage.com	Japan submissions@editage.com +81 03-6888-3348 www.editage.jp	Korea submit- korea@editage.com 1544-9241 www.editage.co.kr	China fabiao@editage.cn 400-005-8055 www.editage.cn	Brazil contato@editage.com 0800-892-20-97 www.editage.com.br
--	--	---	--	---

Introduction

Line 51, page 2: Moreover, the surface...bone formation.

References are needed.

Response : A reference has been added.

Line 4, page 3: Hydroxylated or hydrated...differentiated osteoblast phenotype.

References are needed.

Response : A reference has been added.

Line 18, page 3: The presence of TiO₂ layer...Ti in dental implants.

The reviewer disagrees with the authors. If the authors really believe that this description is right, please, explain it in detail, including proper references.

Response : I have explained it in detail, marked in red. And references have been added.

Line 35, page 3: Micro-arc oxidation (MAO)...of micro-plasma.

References are needed.

Response : A reference has been added.

Line 42, page 3: Firmly adherent porous...surfaces for osseointegration [7, 8].

What is the coating material formed on the surface of Ti by MAO? The reviewer thinks that MAO produces a TiO₂ layer. Why is this layer different from that preventing osseointegration? Just the change of crystallinity is not considered to be the transformation from being biotoxic to being biocompatible.

Response :MAO produces porous and rough TiO₂ coatings. In the current study, the coating in the 9%-2MAO group showed a nanoporous structure; 11%-13MAO consisted of equally distributed micro-sized slots (3–7 μm wide) and nano-sized pores (diameter: 10–300 nm) and showed a typical micro/nano dual structure that mimicked the hierarchical characteristics of bone. 11%-13MAO was beneficial for cell adhesion and differentiation. The TiO₂ layer formed spontaneously on the Ti surface owns good biocompatibility and remarkable corrosion resistance. However, it is complete and dense. This inherent oxide layer results in the growth of fibrous connective tissue, preventing new bone formation on the implant surface, which retards osseointegration between the implant and bone.

Materials and methods

2.1. Specimen preparation

Which grade of cp-Ti was used in this study?

Response : TA-2 . I have added it in the part of “2.1. Specimen preparation”, marked in red.

Results

3.1. Material characterization

Line 46, page 8: The surface of Ti was primarily smooth.

The authors are asked for more detailed description of the cp-Ti surface. Why did the authors think that the Fig. 1A image showed a smooth surface? What does the word, 'smooth,' mean exactly?

Response : I have described the cp-Ti surface more detail, corrected the wording, and marked in red.

Line 4, page 9: The surface chemistry was analyzed by XRD (Fig. 2)

Although XRD has some information in surface chemistry, XRD is usually known to analyze a surface in crystallinity. The authors analyzed the surfaces in phase only, not in chemical elements or in the percentages (atomic or weight) of the elements in this study. Maybe, oxygen is also detected on the titanium surfaces if the authors analyze the surfaces by X-ray photospectroscopy or energy dispersive spectroscopy.

Also, the significant figures in the results are recommended to be consistent.

Response :I have changed “the surface chemistry” to “the surface phase structure”, which was marked in red.

Discussion

Line 21, page 14: Here, we aimed...dental implants therapy.

The micro/nano-hierarchical structure of the authors is not considered to be an optimized biomaterial, based on the results of this study. The authors did not perform the experiments depending on the MAO times or the duty circles. The results were not statistically analyzed when these factors were independent variables. The word, 'optimized,' is inadequate in this study. Further studies are definitely needed.

Response : The word, “optimized”, has been deleted. At the last part of discussion, I added the sentence “Further studies are needed to determine the effect of MAO times or duty circles on the structure of the TiO₂ coating”, marked in red. In our study, what’s more, we would like to screen a suitable structured TiO₂ coating for osteoblast adhesion and differentiation in

dental implants. Thus, we prepared different structural TiO₂ by changing the conditions of MAO. In the future study, we will systematically analyse the effect of independent variable (MAO times or duty circles) on the surface structure of coatings.

Please, describe the limitations of this study. The authors did not perform in vivo experiments.

What is the difference between the authors' study and the previous study of Li YD, et al (Reference No. 10), which includes the in vivo results? The reviewer recommends that the authors should describe improvement of this study, compared with the previous one.

Response : At the last part of discussion, I described the limitations of this study. In the first paragraph of discussion, the difference between our study and the previous study of Li YD, et al has been given. And I discussed improvement of this study. They were marked in red.